# CLEVRTEX: A Texture-Rich Benchmark for Unsupervised Multi-Object Segmentation

**Laurynas Karazija**     **Iro Laina**     **Christian Rupprecht**

**Visual Geometry Group, University of Oxford**
{laurynas, iro, chrisr}@robots.ox.ac.uk

## Abstract

There has been a recent surge in methods that aim to decompose and segment scenes into multiple objects in an unsupervised manner, *i.e.*, unsupervised multi-object segmentation. Performing such a task is a long-standing goal of computer vision, offering to unlock object-level reasoning without requiring dense annotations to train segmentation models. Despite significant progress, current models are developed and trained on visually simple scenes depicting mono-colored objects on plain backgrounds. The natural world, however, is visually complex with confounding aspects such as diverse textures and complicated lighting effects. In this study, we present a new benchmark called CLEVRTEX, designed as the next challenge to compare, evaluate and analyze algorithms. CLEVRTEX features synthetic scenes with diverse shapes, textures and photo-mapped materials, created using physically based rendering techniques. It includes 50k examples depicting 3-10 objects arranged on a background, created using a catalog of 60 materials, and a further test set featuring 10k images created using 25 different materials. We benchmark a large set of recent unsupervised multi-object segmentation models on CLEVRTEX and find all state-of-the-art approaches fail to learn good representations in the textured setting, despite impressive performance on simpler data. We also create variants of the CLEVRTEX dataset, controlling for different aspects of scene complexity, and probe current approaches for individual shortcomings. Dataset and code are available at https://www.robots.ox.ac.uk/~vgg/research/clevrtex.

## 1   Introduction

Supervised scene understanding has seen significant progress in the last decade. The introduction of deep learning to the field and large, manually annotated datasets have made it possible to address tasks such as object detection [39], semantic or instance segmentation [27], layout prediction [57] and dense captioning [31] with considerable accuracy. However, in absence of labels, and thereby supervision, such tasks are exceedingly difficult, even though it is easy to imagine that with enough images (or videos), it should be possible to identify objects and the general composition of a scene without human annotations. This renders unsupervised multi-object segmentation, as well as object-centric learning a challenging yet promising field with high potential.

While certain tasks in the general context of *unsupervised* scene understanding and decomposition have a relatively long history in computer vision, the majority of applications focus on single objects: image classification [8, 29, 53], saliency detection [45, 61], foreground/background segmentation [2, 10, 43, 54] and general image-level representation learning [9, 11, 25, 28]. These methods are usually developed on datasets such as ImageNet [48] that contain one object of interest per image. Nevertheless, most real-world scenes are often comprised of multiple objects in varying spatial configurations.

35th Conference on Neural Information Processing Systems (NeurIPS 2021) Track on Datasets and Benchmarks.

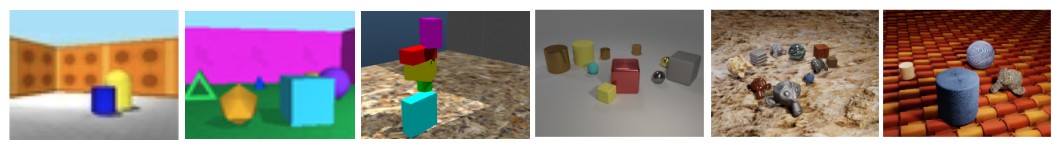

GQN [18]      ObjectRoom [6]  ShapeStacks [26]   CLEVR [32]                CLEVRTEX

Figure 1: Qualitative comparison of our new CLEVRTEX dataset with previous unsupervised multi-object learning datasets featuring 3D objects. See Table 1 for quantitative comparison.

Table 1: Comparison of the proposed CLEVRTEX dataset with previous unsupervised multi-object learning datasets featuring 3D objects.

| Dataset | #Images | #Objects | #Shapes | #Obj. Colors | #Obj. Materials | #Backgrounds | Annotations |
|---|---|---|---|---|---|---|---|
| GQN [18] | 12M | 1–3 | 7 | — | 1 | 15 | Camera parameters |
| ObjectRoom [6] | 1M | 1–6 | 4 | 10 | 1 | 100 | Semantic, factor of variation |
| ShapeStacks [26] | 310k | 2–6 | 4 | 5 | 1 | 25 | Semantic, stability, stability type |
| CLEVR [32] | 100k | 3–10 | 3 | 8 | 2 | 1 | Semantic, factors of variation |
| CLEVRTEX (Ours) | 50k+10k | 3–10 | 4+4 | — | 60+25 | 60+25 | Semantic, depth, normal, shadow, factors of variation |

Only recently, methods have been developed to analyze and decompose whole scenes containing multiple objects, *i.e.*, jointly learning to represent and segment objects from raw image input, *without* supervision. However, since moving from individual objects to complex scenes drastically complicates the problem, these methods currently rely on simple synthetic datasets. The complexity of these datasets ranges from simple, single-color 2D shapes arranged against a black background [6] to rendered 3D scenes composed of uniformly colored, 3D primitives (cubes, spheres, cylinders) [32] (Fig. 1). Interestingly, current methods work *very well* on this kind of data and saturate the existing benchmarks such that a quantitative comparison of models becomes difficult.

How to scale such methods to visually complex real-world data remains an open problem. When analyzing the current state-of-the-art methods and datasets, it becomes clear that there is a strong reliance on simple appearance (*e.g.*, single color, simple shape). For example, Greff et al. [24] identify a tendency of their model to segment by color, and it fails when applied to natural images. In fact, the majority of methods learn semantic objects using similar compositional principles, which exploit statistical advantages in aligning simple scene elements with internal representations. Natural images and the objects therein, however, do not possess strong, consistent colors. Instead, they feature confounding textures, often a mixture of repeating and irregular patterns, which might violate such assumptions.

This work introduces a dataset and benchmark as the next step towards eventually tackling real-world scenarios. We propose CLEVRTEX, a synthetic dataset that consists of *textured* foreground objects and background, unlike existing benchmarks. Interestingly, we find that simply moving from uniformly colored to textured objects poses extreme challenges for current models, and no existing method achieves satisfactory performance. For this reason, we also introduce several variants of our dataset to gradually scale the visual complexity of the scenes and investigate where current algorithms struggle. To probe the generalization capability of models to out-of-distribution scenes, we create additional test sets that contain unseen shapes and materials and camouflaged objects. Together with CLEVRTEX and its variants, we are releasing the code to generate the dataset from scratch. Finally, we find that existing work does not rely on a consistent set of metrics and benchmarks. In an extensive set of experiments, we benchmark the majority[1] of current work on both CLEVR and our newly introduced CLEVRTEX.

## 2   Related Work

Object recognition benchmarks such as PascalVOC [19] or MS COCO [37] have been fundamental to object detection research. However, the current unsupervised multi-object segmentation models are yet unable to handle diverse real-world images featured in such datasets and have relied on visually

---

[1]wherever code was available or could be obtained from the authors

trivial 2D and 3D data. Here, we review datasets and benchmarks used in *unsupervised* multi-object segmentation methods and point out the differences to CLEVRTEX.

**2D Datasets**    Earlier unsupervised multi-object learning approaches were applied to transformed versions of existing 2D datasets, often originally crafted for disentanglement research, such as Shapes [46], variants of MNIST [35]: TexturedMNIST [22] and MultiMNIST [49], as well as the multi-object version of dSprites [42], *i.e.*, Multi-dSprites [6]. Others borrow data from the reinforcement learning community, such as the ATARI game environment [1] or Tetrominoes [4]. However, 2D datasets, whilst valuable for development, do not contain the visual cues and details (*e.g.* shadows and perspective) needed to learn object segmentation that generalizes to real images.

**3D Datasets**    Simple 3D Phong-shaded datasets (Fig. 1) have been crafted for use in the unsupervised multi-object setting. The object-room dataset [6], a multi-object extension of 3D shapes [5], features colored shapes arranged in a room with colored walls. ShapeStacks [26] features stacked, solid-colored primitives on a simple background with a pattern. CLEVR [32], which is most closely related to our work, was introduced as a visual question-answering dataset but has become a popular benchmark in unsupervised scene decomposition as well. It features a set of 3-10 primitive shapes arranged on a gray photo backdrop; objects can have either a rubbery or metallic appearance and one of 8 color tints. CLEVR6 [24] is a filtered version of the CLEVR dataset that includes only up to 6 objects per image. It is often used for training in multi-object representation learning, with the remainder of CLEVR used to test generalization to more crowded scenes [14, 40].

Additional variants of CLEVR have also been generated for other tasks, such as ARROW [30] for exploring scene composition accuracy, and a recursive version in [13] for learning part-whole relationships. Multi-view variations [34, 52] are used for 3D representation learning, and further include new object geometry, such as toys [36] and chairs [59]. However, these datasets feature simple scenes of low visual complexity, with contrasting solid colors present on objects. CLEVRTEX instead contains difficult objects with various materials that include repeating patterns and small details and often blend in rather than stand out from the background.

The main differences in data statistics between CLEVRTEX and commonly used multi-object learning datasets are also summarised in Table 1.

**Unsupervised Multi-Object Segmentation in Natural Scenes**    Some attempts have also been made to scale to natural scenes. Eslami et al. [17] apply the AIR model modified with a 3D rendering engine to infer identities and positions of crockery items on a table, training on simulated data, and evaluating against real-world images. Monnier et al. [44] test their sprite-based method on foreground/background segmentation on the Weizmann Horse dataset [3]. Engelcke et al. [16] apply Genesis-V2 to robotic manipulation datasets, Sketchy and APC [60]. Sketchy [7] features recordings of a robotic arm manipulating solid colored toys, towels, or other small objects on a test table, but it lacks segmentation masks. The APC [60] dataset is used instead for evaluation but only contains a single foreground object. These attempts signal promise that unsupervised multi-object segmentation can eventually scale to diverse real-world images.

**Visual Fidelity in Simulation**    Simulation has always been central to progress in machine/reinforcement learning. However, as usual, the gap between a simulated setting and the ability to generalize to real-world environments is of concern. Several new simulators aim to improve the visual fidelity using photo-mapped environments or artists' compositions [33, 41, 50, 56]. Recently, TWD [20] introduced a rich physics engine and PBR rendering of environments with a library of objects. Similar to our work, the emphasis is partly on increasing visual fidelity while moving away from trivial settings and towards real-world applications. However, RL environments have not seen much use in the unsupervised vision domain due to the often specific nature of the data, egocentric perspective, and temporal dependency.

## 3    CLEVRTEX

We introduce CLEVRTEX, a simulated dataset designed to present the next challenge in unsupervised multi-object learning. It introduces confounding visual aspects such as texture, irregular shapes, and

various materials while maintaining control over scene composition. CLEVRTEX is available under CC-BY license.

## 3.1 Dataset Creation

CLEVRTEX is a much more visually complex extension of CLEVR [32] targeted at multi-object learning. It is procedurally generated using the API of Blender[2], a powerful open-source 3D suite.

At the center of the CLEVRTEX generation process is a catalog of diverse photo-mapped materials[3] ranging from forest floor duff, rocks, brickwork, and tiles to fabrics, metallic weaves, and meshes — a full list of materials is shown in Appendix C.5). To generate each image, we start with a scene containing only a photo backdrop, which will become the background. For viewpoint and lighting diversity, we apply random jitter to the position of the camera and three lights. We then fill the scene with 3 to 10 objects (number sampled uniformly), sampling each object from a set of shapes: a cube, a sphere, a cylinder, and a non-symmetric shape of anthropomorphized monkey head[4] for increased complexity in object silhouettes. Objects are added to the scene one by one by sampling position (continuous, $(x, y) \sim \text{Uniform}(-3, 3)$), scale (discrete, $s \in \{.9, .6, .4\}$), and rotation (continuous, $\theta \sim \text{Uniform}(0, 360)$). If a new object collides with already existing shapes in the scene, the object's transformation is resampled until no collision is found or a maximum number of trials is exceeded, at which the process restarts by removing all objects.

We then sample a material for each object and the background. Using adaptive subdivision, we create material-specific geometry by displacing vertices of the starting shapes. This creates reliefs for simpler materials or distorts shapes, extruding features or introducing holes. The materials use albedo, subsurface scattering, and reflectivity maps to generate detailed visuals. Using physically based rendering ensures appropriately detailed reflections, highlights, and lighting effects. In addition, we generate ground truth segmentation maps through the rendering process and automatically check that no object is fully occluded. In that case, the scene is resampled from scratch. Further figures depicting scene lighting, objects, their scales and deformations are available in Appendix C.5.

The object shapes and placement mimics that of the CLEVR dataset [32] for backward compatibility. We do not generate the question-answering part of the original CLEVR dataset but include full metadata. This means

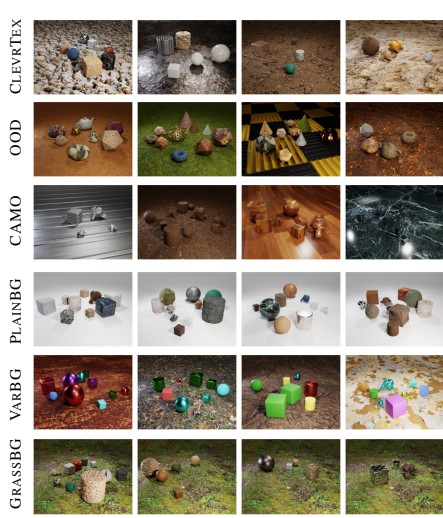

Figure 2: CLEVRTEX and its variants.

that this dataset could also be used for other CLEVR-based tasks such as question answering, although this is not our focus here. Similarly, in anticipation that our dataset might also find usages beyond its intended setting, we include depth, albedo, shadow, and normal maps alongside the images, segmentation maps, and metadata. We share the code to generate CLEVRTEX alongside the dataset.

## 3.2 Statistics

CLEVRTEX contains $50\,000$ images, of which we use 10% for testing, 10% for validation and the remaining 80% ($40\,000$ images) for training. Each image contains between three and ten objects (uniformly sampled). There are four possible shapes, which have been modified to enable clean texture mapping. We use three distinct object scales to maintain identifiable size "names", as in CLEVR, and custom meshes to ensure that the scaling of the objects does not distort texture details. The object placement and rotation are sampled from a continuous range. Note that even though two

---

[2]https://www.blender.org/

[3]We use the computer graphics term "material" to refer to the collection of resources used to creates the likeness of appropriate real-world material on simulated surfaces. Materials are typically a composition of various modalities, such as normal, diffuse, specular, and displacement maps, as well as a computation graph and shaders. We use the term "texture" to refer to 2D images mapping color information onto 3D surfaces.

[4]A modified version of Suzzane – a prefab shape available in Blender.

shapes — cylinder and sphere — are rotationally symmetric, the materials applied to them are not. We use a catalog of 60 materials with non-commercial licenses to generate the whole dataset before splitting the data into training sets. The materials are manually adjusted to ensure visually pleasing results at different scales and the background.

### 3.3 Variants

We create the following modifications of CLEVRTEX, each with 20 000 images (see Fig. 2), to enable a more detailed analysis and evaluation and probe methods for their shortcomings.

The first variant, PLAINBG, is a dataset consisting of textured objects on a plain background, *i.e.*, the background is always set to a simple material as in CLEVR. We also create the reverse version, VARBG (varied background), where the objects are assigned simple CLEVR-like materials and colors while the background receives a textured material at random from our material catalog. PLAINBG and VARBG fall in-between CLEVR and CLEVRTEX in terms of visual complexity. In PLAINBG, intra-object appearance is more complex, but each object clearly stands out from the plain background. On the other hand, VARBG maintains uniformly colored objects but introduces background texture, effectively making the background more diverse than the foreground. PLAINBG and VARBG can be used to analyze the importance of background vs. object reconstruction. Furthermore, we create GRASSBG, which contains scenes with the same mossy grass material as the background, while foreground objects receive materials at random. This variant is thus comparable to CLEVRTEX in terms of visual complexity. However, consistency in the background allows for testing memorization vs. reconstruction effects.

In addition, we propose the following two test sets to serve as an extra check for the limitations of CLEVRTEX.

CAMO contains scenes with "camouflaged" objects. To simulate this, every scene is made of a single, randomly sampled material that is used on all objects *and* the background. CAMO is created to challenge the internal-vs-external consistency and the efficiency hypothesis that underpins compositional methods. The only visual cues here are lighting, shadows, and perspective. It should enable probing models to see if they rely on such context to identify objects. Although we release CAMO with training, validation and test splits, in our experiments it is only used as a testbed for models trained on CLEVRTEX.

Finally, we also provide a separate OOD (out-of-distribution) dataset to evaluate model generalization on novel scenes. This dataset is designed exclusively as a test set and thus only contains 10 000 images. OOD is generated the same way as CLEVRTEX, but exclusively uses 25 *new* (unseen) materials — *i.e.* different from the 60 already used in other variants — and four new shapes (cone, torus, icosahedron, and a teapot) that are not part of CLEVRTEX.

## 4    Models

In recent years, there has been a surge of methods that aim to decompose a scene into objects in an unsupervised manner and, at the same time, learn object-centric representations. Following [38], we categorize these methods as follows.

**Pixel-Space Approaches (⊞)**    A common way to frame the problem of unsupervised scene decomposition into objects is to assign each pixel to one of a usually fixed number of scene components, inferring per-pixel membership maps [6, 14, 22–24, 58]. While these methods are probabilistic in nature, they do not lend themselves to generating new images. For this reason, several generative methods have been proposed, where images can be sampled from the learned distributions [15, 16]. Finally, Locatello et al. [40] introduce a discriminative approach using an iterative clustering-like slot attention mechanism.

Here, we benchmark MONet [6] and IODINE [24] as examples of earlier approaches that handle 3D colored scenes. We also evaluate the improved efficient MORL (eMORL) [14], Genesis-v2 [16] as a generative model, and Slot Attention [40] which is representative for discriminative models.

**Glimpse-Based Methods (▣)**    An alternative to predicting components for each pixel is to extract patches of the input—named *glimpses*—that contain objects. A dense segmentation can be derived in

this reduced space. These glimpses are arranged on top of an explicit background to reconstruct the image. Glimpse-based methods [12, 13, 17, 30, 38] tend to offer computational advantages due to smaller regions, however also require deciding, extracting and composing patches.

From the glimpse-based methods, we benchmark SPAIR [12], which models glimpses auto-regressively, using a truncated geometric prior. Since it cannot handle non-black backgrounds, we modify the model to include a VAE for background prediction (SPAIR*). We also evaluate SPACE [38] due to its use of the pixel-space approach for processing the background, and GNM [30], which uses scene-level priors.

**Sprite-Based Methods (▣)** Recently, several methods [44, 51] propose to decompose images into a learned dictionary of RGBA sprites instead of learning a generative model. From the alpha masks of each sprite, the scene segmentation can be recovered. We benchmark MarioNette [51] and DTISprites [44] to investigate the differences of two sprite-based (▣) approaches.

Table 2: Computational resources for different models. $\times$ indicates number of GPUs needed. Measured on NVIDIA P40 24GB GPUs, with original batch sizes and $128 \times 128$ input. Train. time refers to time required to train the models for the recommended number of iterations, measured in total GPU hours. Inf. time measures the mean inference time required for a single batch, shown $\pm\sigma$ over 7 passes.

| Model | Train. Time (GPU h) | Inf. Time (ms$\pm\sigma$) | Peak GPU Mem (GB) |
|---|---|---|---|
| ▣ GNM [30] | 54 | 258 $\pm9$ | 4 |
| ▣ SPACE [38] | 64 | 191 $\pm2$ | 8 |
| ▣ SPAIR* [12] | 77 | 213 $\pm2$ | 11 |
| ▣ DTI [44] | 198 | 2530 $\pm5$ | 11 |
| ▣ MN [51] | — | — | 11 |
| ⊞ IODINE [24] | $4 \times 202$ | 1360 $\pm2$ | $4 \times 23$ |
| ⊞ SA [40] | 290 | 818 $\pm1$ | 17 |
| ⊞ MONet [6] | $3 \times 106$ | 544 $\pm1$ | $3 \times 17$ |
| ⊞ eMORL [14] | $4 \times 158$ | 217 $\pm1$ | $4 \times 17$ |
| ⊞ GenV2 [16] | 194 | 452 $\pm1$ | 15 |

The aforementioned models have highly varying computational requirements. We offer a side-by-side comparison in Table 2, where computational advantages to glimpse-based methods can be immediately seen, with methods such as GNM and SPACE taking a fraction of time and memory required by even single-GPU pixel-space methods. All implementation details, hyper-parameters, and model changes are reported in Appendix C.3.

# 5 Experiments

**Datasets** We benchmark a wide spectrum of methods using CLEVRTEX and its variants. To test generalization, we evaluate models trained on CLEVRTEX using OOD and CAMO. In addition to our CLEVRTEX and its variants, we conduct experiments on CLEVR to provide a complete side-by-side comparison of methods and the new challenges in CLEVRTEX. All implementation details and preprocessing are reported in Appendix C.1.

**Metrics** The majority of previous work has used the adjusted Rand index on foreground pixels (ARI-FG) only as an evaluation metric. We share concerns with [15, 44] that this metric does not reflect how well objects are localized by the model and whether they are considered part of the background. Thus, we report mean intersection over union (mIoU) instead, as it considers the background. Further discussion and a side-by-side comparison of ARI-FG and mIoU can be found in Appendix C.2. Furthermore, we judge the quality of the reconstruction output of the models using the mean squared error (MSE). For the models trained on CLEVR and CLEVRTEX, we report results on three random seeds, including their standard deviation.

## 5.1 Benchmark

The results for the benchmark are detailed in Table 3 and in Fig. 3. Next, we discuss our findings regarding the ability of models to separate foreground and background, to handle textured scenes, as well as their training stability and generalizability to new scenes.

**Background Segmentation** Pixel-space methods (⊞) show impressive performance on CLEVR compared against glimpse-based approaches (▣) on the foreground (see Fig. 3). However, if we consider the ability to segment the background (mIoU in Table 3), their performance advantage disappears, with SPAIR* performing the best. We attribute this to the tendency of pixel-space models to assign parts of the background to nearby objects. In glimpse-based methods, however, the

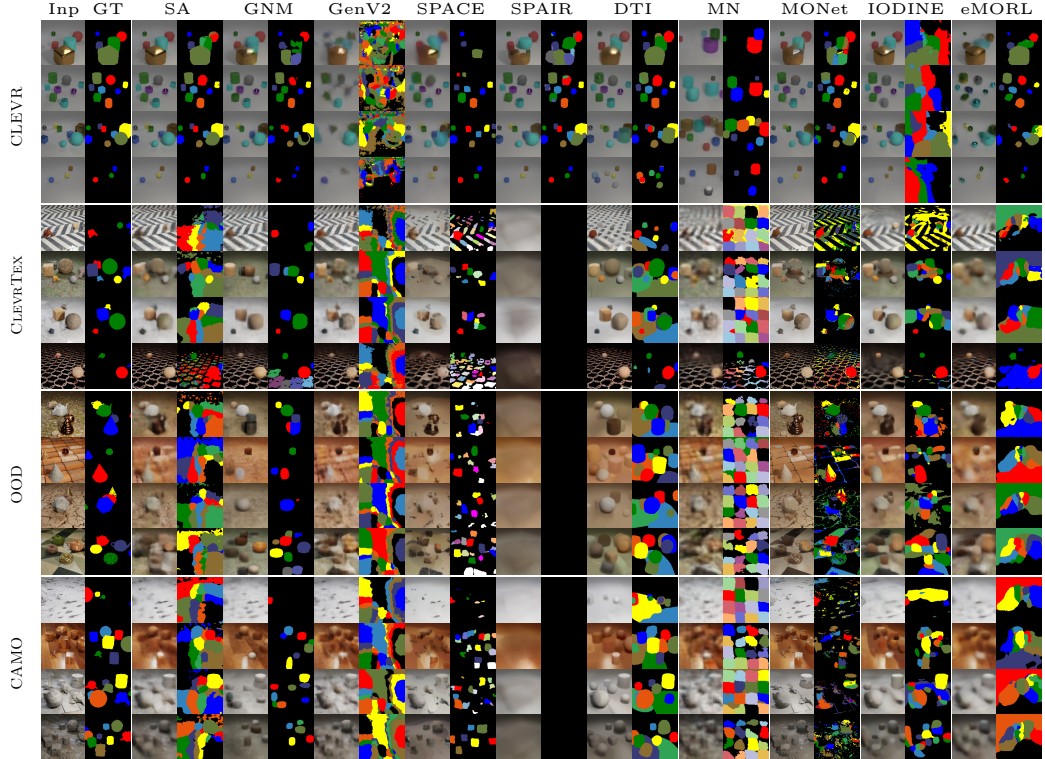

Figure 3: Comparison of various models' reconstruction and segmentation outputs on CLEVR, CLEVRTEX and our test sets. Best viewed digitally. More results in the Appendix, Fig. 5.

Table 3: Benchmark results on CLEVR and CLEVRTEX and the generalization test sets CAMO, and OOD. Results shown $\pm\sigma$ calculated over 3 runs. † updated eMORL: after CLEVRTEX was released, the authors of [14] have updated their codebase to include CLEVRTEX training and evaluation and shared their trained models with improved performance (single seed on CLEVR).

| Model | CLEVR | | CLEVRTEX | | OOD | | CAMO | |
|---|---|---|---|---|---|---|---|---|
| | ↑mIoU (%) | ↓MSE | ↑mIoU (%) | ↓MSE | ↑mIoU (%) | ↓MSE | ↑mIoU (%) | ↓MSE |
| ⊡ SPAIR* [12] | 65.95± 4.02 | 55± 10 | 0.0 ± 0.0 | 1101± 2 | 0.0 ± 0.0 | 1166± 5 | 0.0 ± 0.0 | 668± 3 |
| ⊡ SPACE [38] | 26.31±12.93 | 63± 3 | 9.14± 3.46 | 298± 80 | 6.87± 3.32 | 387± 66 | 8.67± 3.50 | 251± 61 |
| ⊡ GNM [30] | 59.92± 3.72 | 43± 3 | 42.25± 0.18 | 383± 2 | 40.84± 0.30 | 626± 5 | 17.56± 0.74 | 353± 1 |
| ▣ MN [51] | 56.81± 0.40 | 75± 1 | 10.46± 0.10 | 335± 1 | 12.13± 0.19 | 409± 3 | 8.79± 0.15 | 265± 1 |
| ▣ DTI [44] | 48.74± 2.17 | 77± 12 | 33.79± 1.30 | 438± 22 | 32.55± 1.08 | 590± 4 | 27.54± 1.55 | 377± 17 |
| ⊞ GenV2 [16] | 9.48± 0.55 | 158± 2 | 7.93± 1.53 | 315±106 | 8.74± 1.64 | 539±147 | 7.49± 1.67 | 278± 75 |
| ⊞ eMORL [14] | 50.19±22.56 | 33± 8 | 12.58± 2.39 | 318± 43 | 13.17± 2.58 | 471± 51 | 11.56± 2.09 | 269± 31 |
| ⊞ eMORL† [14] | 21.98 | 26 | 30.17± 2.60 | 347± 20 | 25.03± 1.99 | 546± 4 | 19.13± 4.88 | 315± 21 |
| ⊞ MONet [6] | 30.66±14.87 | 58± 12 | 19.78± 1.02 | 146± 7 | 19.30± 0.37 | 231± 7 | 10.52± 0.38 | 112± 7 |
| ⊞ SA [40] | 36.61±24.83 | 23± 3 | 22.58± 2.07 | 254± 8 | 20.98± 1.59 | 487± 16 | 19.83± 1.41 | 215± 7 |
| ⊞ IODINE [24] | 45.14±17.85 | 44± 9 | 29.16± 0.75 | 340± 3 | 26.28± 0.85 | 504± 3 | 17.52± 0.75 | 315± 3 |

formation of glimpses forces the objects to be spatially compact, which offers an advantage when separating the objects from the background.

**Textured Scenes** When training on CLEVRTEX, all models struggle. The foreground segmentation performance reduces, indicating that models fail to assign whole objects to a single component, likely due to the tendency to overfit consistent color regions. The overall segmentation performance is worse as well. MSE is much higher than on CLEVR, with models producing blurry or flat reconstructions, failing to capture much of the rich variation in the input data. SPAIR*, which showed the best overall performance on CLEVR, fails to recognize any objects and instead simply predicts the background. We conjecture that SPAIR's autoregressive handling of objects paired with the use of spatial transformers might make the learning signal too noisy.

Sprite-based models (▣) also perform worse, as the greater variation in appearances is not sufficiently captured by their limited dictionary. While the dictionary size can be increased, the lack of an internal compression mechanism to represent varied appearances will always be a limiting factor in natural world settings. Interestingly, when unable to capture individual objects, MN learns to tile the image with possible color blobs, representing low-frequency information in the image instead. In our tests, similar tiling behavior tends to occur also in glimpse-based models whenever they cannot learn to reconstruct the foreground (see the Appendix, Fig. 6, for examples in other models). Since DTI includes a set of internal transformations, it performs comparatively better on CLEVRTEX.

GNM, a generative glimpse-based approach, has overall the best performance on CLEVRTEX, which we attribute to spatial-locality constraints imposed through the glimpse-based formulation and limited background reconstruction ability due to a simpler background model; *i.e.* comparing to other methods less capacity is spent on the background. Interestingly, GNM shows one of the largest reconstruction errors, despite being the best at scene segmentation, suggesting that ignoring confounding aspects of the scene rather than representing them might aid in the overall task.

Out of the our benchmarked pixel-space methods (⊞), IODINE performs the best in terms of the overall segmentation performance. Our qualitative investigation shows that pixel-space methods that can segment textured scenes largely capture consistent color regions, which occasionally align with objects on scenes with simpler materials. Large patterns in the background or changes in object appearance, often due to lighting result in oversegmentation.

**Stability**    Due to inherent stochasticity in initialization and optimization, one can expect a degree of variation between different model training runs. Many benchmarked models in this study also rely on internal randomness, primarily due to the sampling procedures involved. This influences the learning signal and the configuration the models can learn. Pixel-based approaches and SPACE (which has pixel-space model for background) show higher variance in the performance metrics. Similar to [14, 38, 40], we observe that these methods occasionally fail to use separate components, which causes high fluctuation between different seeds. Glimpse-based methods are more stable with respect to seeds but tend to exhibit higher sensitivity to hyperparameter settings.

**Generalisation**    In addition to benchmarking existing approaches in their ability to learn and handle textured scenes, we are also interested in the degree to which different approaches might rely on specific factors of CLEVRTEX. To this end, we evaluate the models trained on the CLEVRTEX on two additional test sets: CAMO to see whether models rely on the difference of object appearances present in a scene, and OOD to see whether a degree of memorization (*e.g.* of shapes and materials) plays a role in recognition and whether the methods could generalize to unseen patterns.

Interestingly, some of the better performing approaches on CLEVRTEX maintain much of their segmentation ability on out-of-distribution (OOD) data. GNM, for example, attempts to reconstruct the input using memorized training data materials and shapes, which leads to reduced but still comparable object segmentation. Other sprite- (▣) and glimpse-based (▢) methods either do not perform well or show similar reliance on the appearances from the training distribution. Pixel-space models (⊞) show a better ability to reconstruct the input but also tend to reconstruct based on consistent color regions rather than objects, a tendency only exacerbated by the out-of-distribution setting.

When considering the challenging CAMO setting, none of the approaches perform satisfactory segmentation. Methods that somewhat work on CLEVRTEX tend to use different components to represent lighter and darker parts of the scene, highlighting the tendency of all current models to overfit the scene appearance.

## 5.2   Variants

As discussed above, many of the models that perform well on CLEVR, either do not work on CLEVR-TEX or lose much of their performance. To further probe which aspects of the scene composition are challenging, we use the variants of CLEVRTEX.

**Textured Objects**    When applied to PLAINBG, where materials are only seen on objects, and the background is gray, all of the methods still perform worse than on CLEVR, with a significant drop in segmentation performance, especially prevalent in pixel-space approaches (⊞). Since all methods

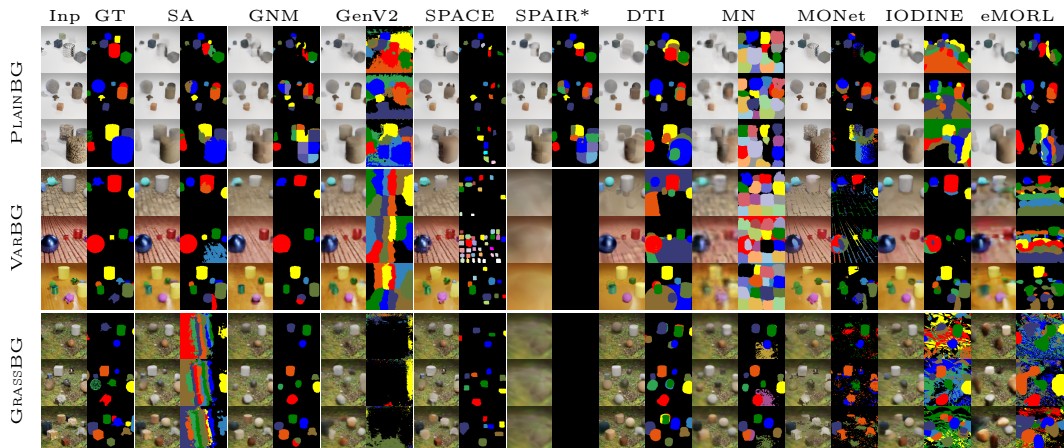

Figure 4: Comparison of various models' reconstruction and segmentation outputs on PLAINBG, VARBG and GRASSBG variants. Best viewed digitally.

Table 4: Model results on PLAINBG, VARBG, and GRASSBG variants.

| Model | PLAINBG | | VARBG | | GRASSBG | |
|---|---|---|---|---|---|---|
| | ↑mIoU (%) | ↓MSE | ↑mIoU (%) | ↓MSE | ↑mIoU (%) | ↓MSE |
| ▣ SPAIR* [12] | 39.32 | 134 | 0.00 | 1246 | 0.00 | 728 |
| ▣ SPACE [38] | 31.96 | 120 | 16.10 | 311 | 33.85 | 196 |
| ▣ GNM [30] | 26.49 | 96 | 49.78 | 438 | 53.15 | 254 |
| ▣ MN [51] | 10.16 | 167 | 11.51 | 441 | 34.80 | 266 |
| ▣ DTI [44] | 36.03 | 210 | 38.82 | 498 | 37.65 | 215 |
| ⊞ GenV2 [16] | 24.39 | 98 | 14.40 | 298 | 2.88 | 306 |
| ⊞ eMORL [14] | 29.39 | 96 | 22.92 | 385 | 19.38 | 199 |
| ⊞ MONet [6] | 38.72 | 83 | 23.73 | 212 | 21.29 | 165 |
| ⊞ SA [40] | 39.32 | 134 | 62.57 | 257 | 12.88 | 116 |
| ⊞ IODINE [24] | 23.83 | 128 | 39.86 | 364 | 25.76 | 225 |

have been designed with simpler datasets and uniformly colored objects, the more realistic nature of the materials in CLEVRTEX poses a difficult challenge. Glimpse-based models (▣) also show reduced segmentation quality over CLEVR. MN (sprite-based) struggles as the increased diversity in foreground objects overwhelms the spite dictionary. Finally, the models' inability to capture the fine-grained details of the more complex object appearance causes the increase in reconstruction error.

**Textured Background** VARBG contains simple mono-colored objects arranged on top of a diverse set of textured backgrounds. Certain models, like SPAIR*, SPACE, and GenV2, struggle to handle diverse backgrounds. Other methods, however, seem to benefit from simpler objects, showing improvements in segmentation performance over both PLAINBG and CLEVRTEX scenarios, indicating that these models rely on simpler, more consistent objects.

**Consistent Background** GRASSBG has the same complex forest grass background in all scenes. The background is richer and more complex than in PLAINBG. As glimpse-space methods (▣) tend to model the background explicitly, we observe that contrasting consistent background aids these models greatly. Pixel-space methods (⊞) also perform slightly better in this setting than on CLEVRTEX where the background varies. However, the effect is not as pronounced as for glimpse-based (▣) approaches, with the overall performance roughly matching what was observed on CLEVRTEX.

## 6 Conclusions

Unsupervised object learning and scene segmentation is a challenging task. Interestingly, given the existing metrics and commonly used datasets (*e.g.*, CLEVR), current approaches show impressive

performance, yet we have shown that they are easily challenged when visual complexity increases. To this end, we present CLEVRTEX, a new benchmark that aims to increase visual scene complexity, which contains richer textures, materials, and shapes, to encourage progress towards methods applicable to real images in the wild.

In our experiments, GNM [30] and IODINE [24] perform the best out of glimpse-based and pixel-space models, respectively, with GNM showing the best segmentation performance overall. However, almost all methods struggle to handle multiple textured scenes, resulting in a significant performance gap with respect to the closest current benchmark, CLEVR. Our findings suggest that pixel-space methods tend to be more prone to overfitting consistent color regions and smooth gradients. On the other hand, sprite- and glimpse-based approaches tend to memorize small repeated patterns, which offers an advantage on CLEVRTEX. Further testing, however, shows that these models reconstruct smooth backgrounds and recognize sharp changes as objects. As such, even the approaches that show some ability to handle textured environments focus largely on scene appearance, failing to learn and exploit global context clues that might align with semantic objects.

We believe that textures pose a challenge to current pixel-space and glimpse-based methods as they are built to exploit simple visual elements and uniform appearance that is present in previous datasets, partly due to the reconstruction objectives. We find evidence for this in our experiments with the dataset variants: consistency within *objects*, as seen in our VARBG variant, and consistency in *backgrounds* (PLAINBG and GRASSBG) helps to learn better models than the full CLEVRTEX where there is no simple intra- and inter-appearance consistency. Only on simpler scenes (Fig. 3) the best performing methods succeed at segmenting some objects.

Thus, CLEVRTEX offers new challenges for unsupervised multi-object segmentation, especially for evaluating generalization. Furthermore, the three variants and two additional test sets can serve as a diagnostic tool for developing new methods, and the extensive evaluation acts as a standardized benchmark for current and future methods.

**Limitations**   The proposed dataset contains a limited number of primitive shapes and a catalog of 60 materials. Although future models might exploit the non-exhaustive nature of object appearance, *e.g.*, memorizing object reconstructions than learning generalizable scene decompositions, we have shown that current methods are, in fact, faced with a significant challenge, even at a slight increase of data complexity (*e.g.*, on PLAINBG). To further address this limitation, we have created the OOD dataset, which should serve as an additional test for the generalization ability of models outside the training distribution. Overall, CLEVRTEX is still a synthetic dataset and does not fully close the gap to real-world data. However, until methods can solve CLEVRTEX, generalization to real images is likely out of reach.

**Broader Impact**   The work presented here critically evaluates current approaches for unsupervised multi-object segmentation. The introduced datasets are fully simulated renderings of 3D primitives and do not contain any people or personal information. Our benchmark aims to establish and standardize evaluation practices, provide new challenges for current algorithms, and help future research compare with prior work. While CLEVRTEX is highly important for current research, its impact outside of the research community is low as current methods can not yet properly deal with real images.

## Acknowledgments and Disclosure of Funding

L. K. is funded by EPSRC Centre for Doctoral Training in Autonomous Intelligent Machines and Systems EP/S024050/1. I. L. is supported by the European Research Council (ERC) grant IDIU-638009 and EPSRC VisualAI EP/T028572/1. C. R. is supported by Innovate UK (project 71653) on behalf of UK Research and Innovation (UKRI) and by the ERC IDIU-638009. We thank Johnson et al. [32] for their open-source implementation of CLEVR. We would also like to thank Martin Engelcke for helpful suggestions on applying Genesis-V2 to CLEVRTEX, Patrick Emami for assistance adapting eMORL to CLEVRTEX and Dmitriy Smirnov for sharing their implementation of MarioNette.

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
