# OpenReview forum: "ClevrTex: A Texture-Rich Benchmark for Unsupervised Multi-Object Segmentation"
_NeurIPS.cc/2021/Track/Datasets_and_Benchmarks/Round2 — NeurIPS 2021 Datasets and Benchmarks Track (Round 2)_

### Official Review · Reviewer_pu3C · 2021-09-20
**Interesting Textured Dataset**

**Rating:** 6
**Confidence:** 3
**Clarity:** I believe so.

**Strengths:**

This paper clearly verifies some of my thoughts in the last few years, that is, many of existing computer vision systems failed to consider visual realism and complexity, especially those that heavily rely on synthetic data. As a computer graphics researcher, I fully understand the importance of having a texture and material rich dataset, therefore I appreciate any effort along this direction.

The variants subsets look interesting to me since I can see authors carefully construct them to test specific capability of existing methods, and I think this is the correct way of decomposing the dataset into more useful parts.

**Weaknesses:**

Although I do like the idea of increasing the visual complexity of multi-object dataset, there are some weakness of this work:
1. The dataset diversity is still questionable in terms of material, texture, shape, and lighting. Currently there are only 60 materials, a few base shapes and texture maps, and a few different lighting conditions used to create the dataset. I would recommend using Adobe's material library which contains much more realistic ones. It is always a good idea to apply some levels of bumping on the surface to further increase the variation. As for lighting, I also recommend using some environment light datasets that are currently available to enrich the random area lights.
2. There is no new methods proposed in this work to address some challenges of the new dataset, so the technical contribution is another weak point of this paper.
3. The paper is written in a way that makes me think it is only useful for object segmentation. However when I took a look at the dataset there are more auxiliary channels such as albedo, depth, normal, etc. So this looks to me can be also used for other tasks such as depth or orientation estimation for robotics tasks. Only considering a single task however weakens the usefulness of the dataset.

**Additional Feedback:**

My suggestions for improvement are included in the weakness comment.

**Correctness:**

I believe the contents are correct, since both dataset and code are available online to allow anyone to check.

**Documentation:**

Yes, there are sufficient details.

**Ethics:**

There are no concerns.

**Relation To Prior Work:**

I think the authors did a good job on discussing about previous works and datasets by showing some pictures that contain noticeable differences.

**Summary And Contributions:**

This paper presents a new texture-rich dataset that can be used for object level 3D understanding such as segmentation. Compared to existing 2D and 3D datasets, this one contains higher visual complexity and is hard for current methods to perform very well on it.

---

> ### Author Response · Authors · 2021-09-28
> **Response**
>
> Thank you for the careful consideration, thoughtful feedback and suggestions for improvement.
>
> > The dataset diversity is still questionable in terms of material, texture, shape, and lighting. Currently there are only 60 materials, a few base shapes and texture maps, and a few different lighting conditions used to create the dataset.
>
> As we state in Sec. 6, we agree that the catalog of shapes and materials could be expanded. However, current algorithms already struggle to achieve any satisfactory results, as visual scene complexity is significantly increased in comparison to the datasets that have been used to date. To provide a reasonable next challenge for research, a benchmark should not be too difficult. To address the potential problem where models circumvent the task and memorise the limited catalog, thereby segmenting the scenes by texture alone, we have designed the out-of-distribution test set (OOD), which features different shapes and 25 additional (novel) materials. As the dataset generation code will be published alongside the paper, a future extension is easily possible.
>
> We have also updated the Appendix to include figures for the lighting variation and shape perturbations used in the generation process (Fig. 7 and 8).
>
> > I would recommend using Adobe's material library which contains much more realistic ones. It is always a good idea to apply some levels of bumping on the surface to further increase the variation. As for lighting, I also recommend using some environment light datasets that are currently available to enrich the random area lights.
>
> Thank you for the suggestion! Our primary motivation for selecting our current source of materials has been to ensure they are freely available to other researchers who might want to recreate or adjust the data. As far as we can tell, Adobe’s library is limited in the number of assets offered with suitable licenses but we will take it and other texture and environment datasets into consideration for future dataset versions and extensions.
>
> >There is no new methods proposed in this work to address some challenges of the new dataset, so the technical contribution is another weak point of this paper.
>
> As this is a submission for the dataset track, we do not propose a novel method in this paper. We concentrate on designing a benchmark and ensuring extensive evaluation and investigation of the current art in the new challenging setting. We provide side by side comparison, delineate current drawbacks and propose a new challenge to the field.
>
> > The paper is written in a way that makes me think it is only useful for object segmentation. However when I took a look at the dataset there are more auxiliary channels such as albedo, depth, normal, etc. So this looks to me can be also used for other tasks such as depth or orientation estimation for robotics tasks. Only considering a single task however weakens the usefulness of the dataset.
>
> We constructed ClevrTex for use in the unsupervised multi-object segmentation setting to investigate and probe methods for their weaknesses. We are delighted that the reviewer also sees the use of this dataset, and we certainly invite its adoption elsewhere (as already noted in line 155). Similar to CLEVR, which was originally designed for visual question answering, the proposed dataset can potentially be adopted in many communities. However, a comprehensive benchmark and evaluation of this dataset in several different fields is beyond the scope of a single paper.

---

### Official Review · Reviewer_u9BT · 2021-09-21
**Useful Benchmark, minor details missing**

**Rating:** 7
**Confidence:** 4
**Correctness:** The construction and analysis seem so…

**Strengths:**

The dataset is well-motivated and is a timely contribution to spur on new research in unsupervised scene understanding. Extensive analysis of existing methods when trained on this data and tested on several variants is convincing and a valuable contribution as well. Additional ground truth data can also make the dataset useful beyond the applications described as the primary motivation.

**Weaknesses:**

The dataset construction pipeline should be described more precisely. Specifically:

- L 120: How were the photo backdrops selected, how many were used, and what were the splits for train/test?
- Section 3.1: range of the number of objects per scene; camera parameters, scales.
- Section 3.2: more details on material splits.
- Tb 1: it is not clear what the total time is for, is it training, inference, what is the input...?
- Supplemental: include all 60 materials used; include lighting variation demonstration; include object shape perturbation examples.


**Additional Feedback:**

This is useful work that will benefit the community.

**Clarity:**

The paper is clear with the exception of a few missing details. The abstract should list important details about the dataset, including number of images, number of materials, variations, as well as additional ground truth available -- this will make it easier for readers to asses if the data is suitable for their purpose.

**Documentation:**

Code will be made available. In addition, clear licensing terms should be included, and the key details mentioned above.

**Ethics:**

No concerns.

**Relation To Prior Work:**

The key difference is clear. However, this should be more clearly presented. A table summarizing differences with Clevr and any other relevant datasets should be included for clarity, including number of images, number of backdrops, number of object classes, number of materials, range for the number of objects per scene, ground truth annotations.

**Summary And Contributions:**

The paper presents a synthetic dataset with a similar make up to Clevr (Johnson et al 2017), but visually much more challenging, with diverse lighting, textures and object shape perturbations. The principal motivation for the benchmark, is creating a more challenging test ground for unsupervised scene understanding and decomposition methods, and the paper presents extensive analysis that shows that the variation in textures causes existing methods' performance to degrade drastically. The dataset is well-motivated and will be a useful contribution to enable research in this area.

---

> ### Author Response · Authors · 2021-09-28
> **Response**
>
> We thank the reviewer for their kind comments and thoughtful suggestions. We have included the requested details in the revised paper (Sec. 3 and C.5). Here, we will further comment and summarise the changes made.
>
> >How were the photo backdrops selected, how many were used, and what were the splits for train/test?
>
> There is only a single photo-backdrop shape, which has different materials applied and is rendered as a background in each image. We sample material for the background at random, same as for objects, from the catalog of 60 materials for the main dataset and 25 materials for the OOD variant. The minor difference between applying materials to the objects or the background is in scaling. Background material needs to be rescaled to ensure an appropriate level of detail. We have updated the paper (Sec. 3.1) to ensure this is stated more clearly.
> As stated in Sec. 3.2, we split the data 10%/10%/80% for test, validation and training. As we outline in the datasheet for datasets section A.2, these splits are done based on the example index (first 10% or 0-4999 is test, 5000-9999 is validation, and the rest is training). This simple scheme is possible due to the independent random generation of each sample. Each variant has its own splits.
>
> > range of the number of objects per scene; camera parameters, scales.
>
> There are 3-10 objects per scene, a perspective camera of 0.035 focal length and 0 shift. We have also added these details when they are first mentioned (Sec. 3.1) and included the specification of camera intrinsics in the supplementary materials (Sec. C.5).
>
> >Section 3.2: more details on material splits.
>
> We use 60 materials in the ClevrTex, PlainBG, VarBG, GrassBG, and CAMO variants. The datasets are split into test/val/train after generation, and each split contains instances of all material types in roughly the same proportions. We use a disjoint set of 25 new materials and 4 new shapes to generate a test-only OOD variant. We have improved our wording across Sections 3.2 and 3.3 to clarify this and added further details to Sec. C.5 (Fig. 7).
>
> >Tb 1: it is not clear what the total time is for, is it training, inference, what is the input...?
>
> “GPU h” is the time to train a single model for the recommended number of iterations using the 128 x 128 input of recommended batch size. We have updated the table to clarify this information. Our main aim with this information was to point out the sometimes surprisingly large amount of compute that is necessary. We will also include inference times in the final version.
>
> > Supplemental: include all 60 materials used; include lighting variation demonstration; include object shape perturbation examples.
>
> We have included additional material (Sec. C.5, Fig. 7-11). Thank you for the suggestion.
>
> >The key difference is clear. However, this should be more clearly presented. A table summarizing differences with Clevr and any other relevant datasets should be included for clarity, including number of images, number of backdrops, number of object classes, number of materials, range for the number of objects per scene, ground truth annotations.
>
> We have included the table (Tab. 1, revised paper) listing quantitative differences alongside our qualitative picture (Figure 1).
>
> We have also updated the abstract and added the license (CC-BY) to the main text.

---

### Official Review · Reviewer_SsSv · 2021-09-21
**A Good Benchmark for Unsupervised Multi-Object Segmentation**

**Rating:** 7
**Confidence:** 3

**Strengths:**

+ The dataset is well motivated and well designed. Its utility is demonstrated through a carefully designed set of experiments. The experiments cover all aspects of dataset variants one can possibly think of.

**Weaknesses:**

- It might be better to provide some (high-level) insights into why texture poses a challenge to state-of-the-art methods

**Additional Feedback:**

- See comments above.

=============================
Final comments: I am happy with the answers from authors. I keep my original rating.

**Clarity:**

- Very well written.

**Correctness:**

- Correct.

**Documentation:**

- Yes, the code is promised to be released.

**Ethics:**

- No ethical issues.

**Relation To Prior Work:**

- Clearly discussed.

**Summary And Contributions:**

This paper presents ClevrTex, a textured benchmark for Unsupervised Multi-Object Segmentation. It is synthesized using similar procedures as Clevr, but differs in that it adopts a textured background and textured objects. Its goal is to test the current arts' ability to work on textured objects instead of textureless objects, and experimentally shows that texture poses a huge challenge to existing arts. Extensive studies are performed on a varieties of dataset variants, showing the necessity of this dataset as a challenging benchmark to evaluate Unsupervised Multi-Object Segmentation.

---

> ### Author Response · Authors · 2021-09-28
> **Response**
>
> We thank the reviewer for the positive feedback and comments.
> >It might be better to provide some (high-level) insights into why texture poses a challenge to state-of-the-art methods
>
> It is very difficult to pinpoint one specific reason for all methods.
>
> We believe that textures pose a challenge to current pixel-space and glimpse-based methods as they are built to exploit simple visual elements and uniform appearance that is present in previous datasets, partly due to the reconstruction objectives. We find evidence for this in our experiments with the dataset variants: consistency within _objects_, as seen in our VarBG variant, and consistency in _backgrounds_ (PlainBG and GrassBG) helps to learn better models than the full CleverTex where there is no simple intra- and inter-appearance consistency. Large variation in appearances is also challenging for sprite-based methods, as they are limited by their dictionary size (Sec 5.1, Textures scenes).
>
> We have updated Sec. 6 to discuss these insights in more detail.

---

### Author Response · Authors · 2021-09-28
**Overall Response**

We thank the reviewers again for their time and positive feedback. We have addressed comments individually and have uploaded a revised version of the paper. We have highlighted any additions to the text in purple. The following is the summary of changes:
 - We have included additional seeds for larger model IODINES, EMORL and MONet.
 - We have updated several table captions and included further details in the main text.
 - We have included an additional section in the supplementary material to showcase shape, size, lighting diversity, materials and bring additional clarity to the data generation procedure.
 - We made a minor change to the abstract to list key details about the dataset as suggested by reviewer u9BT.

---

### Decision · Program_Chairs · 2021-10-09

**Decision:**

Accept

**Comment:**

All reviewers recommend acceptance and point out the good execution and potential impact of the work in the community. The AC has no reason to overturn the reviewers' decision. Congratulations to authors!